Are improper kinetic models hampering drug development?

Walsh Ryan rwalsh@connect.carleton.ca
Department of Chemistry, Carleton University , Ottawa, ON , Canada
Tulkens Paul
Electronic publication date: 2014 Oct 28
Publication date: 2014
Volume: 2
Electronic Location ID: e649
Received 2014 Sep 12; Accepted 2014 Oct 12
Copyright: © 2014 Walsh
Copyright year: 2014
Copyright holder: Walsh
License: This is an open access article distributed under the terms of the Creative Commons Attribution License, which permits unrestricted use, distribution, reproduction and adaptation in any medium and for any purpose provided that it is properly attributed. For attribution, the original author(s), title, publication source (PeerJ) and either DOI or URL of the article must be cited.
License URL: https://creativecommons.org/licenses/by/4.0/

Keywords: Enzyme kinetics, Disease, Alzheimer’s disease, Drug development, Gamma-secretase, Empirical models, Amyloid, Enzyme inhibition, Irreproducibility, Reproducibility

Funding: The author declares there was no funding for this work.

==============================
Reproducibility of biological data is a significant problem in research today. One potential contributor to this, which has received little attention, is the over complication of enzyme kinetic inhibition models. The over complication of inhibitory models stems from the common use of the inhibitory term (1 + [I]/Ki), an equilibrium binding term that does not distinguish between inhibitor binding and inhibitory effect. Since its initial appearance in the literature, around a century ago, the perceived mechanistic methods used in its production have spurred countless inhibitory equations. These equations are overly complex and are seldom compared to each other, which has destroyed their usefulness resulting in the proliferation and regulatory acceptance of simpler models such as IC50s for drug characterization. However, empirical analysis of inhibitory data recognizing the clear distinctions between inhibitor binding and inhibitory effect can produce simple logical inhibition models. In contrast to the common divergent practice of generating new inhibitory models for every inhibitory situation that presents itself. The empirical approach to inhibition modeling presented here is broadly applicable allowing easy comparison and rational analysis of drug interactions. To demonstrate this, a simple kinetic model of DAPT, a compound that both activates and inhibits γ-secretase is examined using excel. The empirical kinetic method described here provides an improved way of probing disease mechanisms, expanding the investigation of possible therapeutic interventions.

The Problem with Classical Inhibition Models

Inhibitors bind to enzymes according to the same principles that govern ligand and receptor interactions. That is enzyme inhibitors are subject to the same mass action kinetic principles used to define the Hill–Langmuir equation and the Michaelis–Menten equation. However, the way enzyme inhibition equations are currently produced suggests that kinetically enzyme inhibitor interactions are as unique as the enzyme inhibitor system they are used to represent. This problem stems from the supposed mechanistic derivation of the principle inhibition equations. Mechanistic approaches resulted in an ambiguous inhibitory term that does not distinguish between the stoichiometric enzyme inhibitor binding interactions, defined by the inhibition constant (ki) and the effect the inhibitor has on the enzyme.

The principal inhibition equations competitive, non-competitive and mixed non-competitive inhibition were produced using the reaction schemes depicted in Fig. 1 (McElroy, 1947). As can be observed in the competitive inhibition reaction scheme (Fig. 1A), the equation results from the blockade of enzyme substrate interactions by the inhibitor. This equation has been exclusively used to describe a blockage produced by inhibitor binding to the active site of the enzyme. Alternatively the non-competitive inhibition equation (Fig. 1B) is derived from a reaction scheme where the inhibitor binds the enzyme substrate complex. The mixed non-competitive inhibition equation (Fig. 1C) is used to describe inhibitors that can bind to the free enzyme or the enzyme substrate complex.

Figure 1 Classic inhibition schemes.

The (A) Competitive (B) Noncompetitive and (C) Mixed-noncompetitive equations and the reaction schemes used to derive them. In the reaction schemes E represents enzyme; S, substrate; I, inhibitor and P, the product. In the equations v, is equal to the velocity of the reaction; V1, is the maximum velocity usually denoted as Vmax; K1, the substrate binding constant commonly denoted as the Km (Michaelis–Menten substrate affinity constant); Ki, the inhibitor binding constant.

When modeling the competitive inhibition equation an exclusive decrease in the substrate affinity (increase in K1 value) is observed. In contrast, the non-competitive equation produces an exclusive decrease in the maximum reaction rate (V1), and the mixed non-competitive inhibition equation describes inhibition where both substrate affinity and reaction rate are affected.

The flaws in the assumptions used to generate these inhibitory models are easy to identify and has resulted in a field of scientific inquiry continually producing additional equations, to fill in the gaps. For example, changes in substrate affinity can result from mechanisms other than the inhibitor binding to the active site. Kinetic characterization of mutant enzymes or the comparisons of enzymes from different species has clearly demonstrated that substrate affinity directly relates to the spatial conformation of the active site. So competitive inhibition that is believed to be the predominant way in which inhibitors affect substrate affinity is not able to account for inhibitor interactions that alter the spatial conformation of the active site. An example of this false competitive inhibition is the inhibition of the peptidase kallikrein by benzamidine (Sousa et al., 2001). Inhibition of kallikrein by benzamidine is known to occur when benzamidine blocks the binding of peptide side chains but not the active site (Bernett et al., 2002). This results in a decrease in the peptidases substrate affinity but not catalytic activity, which can be conceptualized as the production of a new inhibitor derived surface which is less favorable for substrate binding (Walsh, Martin & Darvesh, 2011; Walsh, 2012).

The assumptions used to generate the non-competitive model only take into account complete catalytic inhibition. A schematic representation of this is usually depicted as an inhibitor binding to the enzyme in a location other than the active site (Fig. 1), inducing a change that prevents substrate hydrolysis. The main problem with this schematic representation is there is no way to distinguish kinetically between non-competitive and competitive inhibition using these models. In each case inhibitor binding would drive product formation to zero as inhibitors binding at the active site or in a peripheral site would prevent enzymatic catalysis. Additionally the non-competitive inhibition model fails to consider situations where the inhibitor may only partially reduce the catalytic rate of the enzyme. Such effects can easily be rationalized by considering situations where the inhibitor binds and does not affect the substrate binding site. This binding may however alter the spatial conformation of the amino acids or cofactors involved in the catalytic activity, reducing the efficiency of the enzymatic activity.

The mixed non-competitive model, as a combination of the competitive and non-competitive models, describes inhibition that affects substrate affinity and the catalytic rate of the enzyme. However, based on the reaction schemes used to generate the competitive and non-competitive inhibition models both one to one enzyme inhibitor binding events are already accounted for (Fig. 1). To rationalize this problem, the classical mechanistic theory allows mixed non-competitive inhibitor to bind to their target with two different inhibition constants (Ki and αKi). This unusual theory excludes the possibility an inhibitor could affect both substrate affinity and catalytic activity in one enzyme inhibitor binding interaction.

The complexity only continues to grow from there, with the uncompetitive inhibition equation that describes inhibitors that produce linear changes in both substrate affinities and catalytic rate (Dodgson, Spencer & Williams, 1956). This unusual form of inhibition increases the substrate affinity of the enzyme while decreasing activity and has been suggested to result from interactions with multienzyme systems. Eventually, problems with these complete inhibitor models began to be recognized, and a partial inhibition class of models occurred (Segel, 1975; Yoshino, 1987). This complexity did not just affect the single substrate, single inhibitor framework but was replicated in the development of multi-substrate and multi-inhibitor frameworks (Segel, 1975). With the ever expanding base of mechanistic equations and the almost unanimous ambivalence to them expressed by researcher, industry and regulatory bodies, enzyme inhibitory characterization has diminished to the basic inhibitory equations or IC50s. In response to this marginalization of mechanistic kinetic studies, there have been attempts to reformulate the single substrate/inhibitor equations (Fontes, Ribeiro & Sillero, 2000). Unfortunately, they have continued to embrace the same inhibitory terms that do not distinguish between binding constants and inhibitory effect.

One of the reasons that these models have endured for so long may be related to the false sense of security; the equations for these models provide. The common form of the inhibitory term (1 + [I]/Ki) found in every equation suggests that it is working in a similar way. This notion is entirely incorrect. The inhibitory term in the competitive inhibition equation (Fig. 1A) directly affects the substrate affinity (K1) by multiplying into it. The inhibitory term in the non-competitive inhibition equation (Fig. 1B) inversely affects the maximum catalytic activity (V1) by dividing into it. A rearrangement of the non-competitive inhibition equation demonstrates that the described decrease in catalytic activity is directly dependent on an inhibitor binding term that is identical to the mass action terms used to describe, ligand receptor interactions in the Hill–Langmuir equation and substrate enzyme interactions in the Michaelis–Menten equation (Eq. (1)). (1) v=V1SS+K11+IKi=SS+K1V1×1−II+Ki=SS+K1×V1−V1II+Ki

The implications of this are that the non-competitive inhibition equation is the only one of these three equations that is even somewhat correct. A similar rearrangement of the competitive inhibition inhibitory term demonstrates that the change in substrate affinity is not directly related to inhibitor binding. The change in substrate affinity (K1) results from the substrate affinity term being divided by the percent of the enzyme population that is not interacting with the inhibitor (Eq. (2)). This inversion of the inhibitor mass binding term is very similar to the inversion of the Michaelis–Menten equation used in Lineweaver–Burk plots. This inversion linearizes the Michaelis–Menten equation and provides a graphical method of determining enzyme kinetic constants. This linear relationship explains why the competitive inhibition equation can only be used to describe linear increases in the substrate affinity. It also provides the rational as to why the competitive inhibition equation is not useful for describing systems where the inhibitor produces a finite hyperbolic shift in the substrate affinity. (2) v=V1SS+K11+IKi=V1SS+K11−II+Ki

The linear relationship of the competitive inhibition equation and hyperbolic relationship described by the non-competitive equation also suggests an explanation for the use of two binding constants in the mixed non-competitive inhibition equation. Since the inhibitory terms of these equations are affecting the constants of the Michaelis–Menten equation in different ways, two inhibitor binding terms are required Ki and αKi (Fig. 1) (Walsh, 2012).

By tying inhibitory effects to mass action inhibitor binding terms, all three base inhibitory equations and equations derived to explain situations they fail to cover, can be replaced with a simple empirical relationship (Eq. (3)) (Walsh, Martin & Darvesh, 2011; Walsh, 2012). (3) v=SS+K1−K1−K1iII+KiV1−V1−V1iII+Ki

That is, virtually any inhibitor and enzyme interaction can be described as a shift in the catalytic activity that is directly dependent on the mass binding of the inhibitor molecules to the enzyme population being studied (Walsh, Martin & Darvesh, 2011). The change induced by the inhibitor can relate to substrate affinity (K1 → K1i), maximum reaction velocity (V1 → V1i) or both. This distinction between binding and effect provides an easy way of describing the vast spectrum of changes in enzymatic activity that can result from the influence of inhibitors or activators. The modularity produced by describing inhibition in this fashion also provides a base equation that can be expanded to explain more complex kinetic interactions.

Empirical Modeling of Complex Inhibition Kinetics

By recognizing that changes in enzyme kinetic parameters are linked to the mass binding of the inhibitor to the enzyme population, more complex situations can be rationalized and modeled in a simple logical fashion. For example, one of the current targets for Alzheimer’s disease treatment is γ-secretase, an enzyme that cleaves the C terminal of the amyloid precursor protein. This cleavage leads to the production of (1–X) amyloid fragments which results in the formation of the β-amyloid peptides (1–40) and (1–42) (Burton et al., 2008). The formation of β-amyloid (1–42) is highly associated with the pathology of Alzheimer’s disease (Burton et al., 2008). Cellular models of amyloid production, usually, rely on the expression of the 99-aminoacid C-terminal fragment of amyloid precursor protein, known as C99, as a model substrate for γ-secretase. One of the drug candidates that had been investigated using this cellular model is DAPT (N-[N-(3,5-difluorophenacetyl)-L-alanyl]-S-phenylglycine t-butyl ester). DAPT is described as having complex kinetic interactions with γ-secretase. At low amyloid precursor protein concentrations, low concentrations of DAPT have been observed to increase γ-secretase catalytic production of amyloid-β 1–40 and 1–42 (EC50 72 nM) (Svedruzic, Popovic & Sendula-Jengic, 2013). However as DAPT concentrations are increased the activation turns into inhibition (IC50 approx 140 nM) (Burton et al., 2008; Svedruzic, Popovic & Sendula-Jengic, 2013). At higher concentrations of substrate DAPT only appears to inhibit the enzyme (IC50 approx 20 nM) (Burton et al., 2008). These observations are further complicated by the substrate activation and inhibition of γ-secretase by amyloid precursor protein (Svedruzic, Popovic & Sendula-Jengic, 2013). However, these observations are all that is needed to produce a quick, simple empirical model of this interaction. The fact that DAPT produces different effects at different amyloid precursor protein concentrations indicates that the enzyme can be considered to have various forms, at least one at low substrate concentration and one at higher concentrations. The presence of multiple conformations is also supported by the apparent substrate activation of the enzyme at elevated substrate concentrations and subsequent substrate inhibition as the substrate concentration is further increased.

So to start with the substrate interactions can be modeled with an expansion of the Michaelis–Menten equation that accounts for a change in the enzyme activity over increasing substrate concentrations (Eq. (4)) (Walsh, 2012). (4) v=V1SS+K1−V1SS+K2+V2SS+K2−V2SS+K3

At low substrate concentrations the substrate binding (K1) can be described using a term that mimics the Michaelis–Menten equation. At higher substrate concentrations the binding of an additional substrate molecule (K2) pushed the γ-secretase to an increased rate of substrate hydrolysis (V2). At even higher substrate concentrations (K3) the γ-secretase activity is inhibited by the substrate. However, fitting this equation (for examples see Kemmer & Keller, 2010) to the data clearly indicates that the changes in catalytic activity are too extreme to fit with the hyperbolic Michaelis–Menten like curves. By adding Hill coefficients, a more sigmoidal change in the data, induced by the substrate binding, can be modeled (Eq. (5)). (5) v=V1SS+K1−V1SH1SH1+K2H1+V2SH1SH1+K2H1−V2SH2SH2+K3H2

With the enzyme described as different forms relating to substrate concentrations, interactions with DAPT are readily defined. At high concentrations of substrate, DAPT acts as a regular inhibitor of γ-secretase catalytic activity. Inhibitory changes in catalytic activity directly relate to the fraction of the enzyme population bound by inhibitor as described by the mass action binding term. The effect on the substrate affinity and catalytic activity are both dependent on the same binding term, and the inhibition mirrors the inhibition produced by very high concentrations of substrate (Eqs. (6) and (7)). The similarity between these interactions suggests that the inhibition by substrate at higher concentrations may relate to additional substrate interactions with γ-secretase rather than through substrate aggregation. (6) V2=V2−V2−V2iIHxxIHxx+Kxx1Hxx

(7) K2=K2−K2−K2iIHxxIHxx+Kxx1Hxx

The effect on γ-secretase at low concentrations of amyloid precursor protein is only slightly more complicated as it has to be described using two interactions between the enzyme and DAPT. In an expansion of the inhibitory term, (Eqs. (8) and (9)), almost identical to the expansion of the Michaelis–Menten equation above (Eq. (4)), the stimulation at low concentrations of DAPT (V1is, K1is) and inhibition at high DAPT concentrations (V1ii, K1ii) are both described mathematically using mass action terms. (8) V1=V1−V1−V1isIHx1IHx1+Kx1Hx1+V1−V1isIHx2IHx2+Kx2Hx2−V1−V1iiIHx2IHx2+Kx2Hx2

(9) K1=K1−K1−K1isIHx1IHx1+Kx1Hx1+K1−K1isIHx2IHx2+Kx2Hx2−K1−K1iiIHx2IHx2+Kx2Hx2

So empirically, without taking into account any information about the mechanism, a kinetic equation which describes three enzyme states produced by substrate concentration and three interactions between DAPT and γ-secretase can be described (Figs. 2 and 3). This equation allows for the simple evaluation of DAPT as an inhibitor and provides insight into problems DAPT may induce by increasing β-amyloid production at normal amyloid precursor protein concentrations. This empirical approach also provides some perspective on the IC50 and EC50 values (Burton et al., 2008; Svedruzic, Popovic & Sendula-Jengic, 2013) described in the literature. At high concentrations of substrate, DAPT is reported to have an IC50 around 20 nM, which would suggest the DAPT is interacting with forms of γ-secretase predominately bound by two substrate molecule (Kxx1 70 nM). Elevated concentrations of substrate would occur in systems setup to over express substrate. While the disassociation constant between the enzyme and inhibitor is somewhat higher than the IC50 this inhibitory effect may be amplified by the inhibitory effects of the third substrate binding interaction (K3 605 nM expression plasmid). At lower substrate levels DAPT (IC50 140 nM) would be interacting with a γ-secretase population fluctuating between enzymes interacting with a single substrate (Kx2 553 nM) or two (Kxx1 70 nM) producing an apparent higher IC50 resulting from the mix. Additionally the EC50 for activation (72 nM) falls very close to the disassociation constant for DAPT induced stimulation (Kx1 30 nM). Using these models allows for facile hypothesis generation, such as, a dimmer of DAPT may be a practical way to eliminate the activation of γ-secretase produced by lower concentrations of DAPT.

Figure 2 Modulation of γ-secretase by DAPT.

Fitting of (A) the mechanistic equation (Eq. (10)), (B) Equation 10 refit and (C) the proposed empirical equation (Eq. (5)) to the raw data for DAPT and γ-secretase interactions. Each line represents a different concentrations of the amyloid precursor protein C-terminal fragment 99, expression vector (Svedruzic, Popovic & Sendula-Jengic, 2013).

Figure 3 Schematic of the interactions between γ-secretase, its substrate APP and DAPT.

The catalytic hydrolysis of APP is controlled by the number of molecules interacting with γ-secretase. Secondary binding of APP or DAPT increases the potential hydrolytic rate dramatically. However, interactions of a third APP or DAPT molecule shuts γ-secretase off suggesting the enzyme may become clogged or be highly regulated catalytically.

A comparison of the work involved in producing the conventional mechanistic model (Eq. (10)) (Svedruzic, Popovic & Sendula-Jengic, 2013) vs. the empirical approach described here also highlights the utility of this approach. Equation (5) and the expansions used to describe the interactions of DAPT with γ-secretase were produced, analyzed and fit to the data (Supplemental Information 1) in a few hours using the solver feature of Excel (Kemmer & Keller, 2010). The classical mechanistic approach used to describe the same kinetic process involved the generation of a complex reaction schematic with 14 enzyme, substrate and inhibitor interactions (Svedruzic, Popovic & Sendula-Jengic, 2013). This reaction scheme was used to define around 25 disassociation constants and three rate constants. The equation derived from this structure was constructed from a connection matrix which was then fed into Mathematica to produce a simplified version that ultimately only had five kinetic constants, (Eq. (10)). While Eq. (10) does contain fewer parameters than the 17 constants used in Eq. (5), a comparison of the predicted values produced using Eq. (10) with the observed experimental data suggests that Eq. (10) does not fit the data very well (Fig. 2). Refitting the parameters of Eq. (10) only marginally improved the model’s ability to fit the observed data (Fig. 4). (10) v=V1S11+SKsi11+IKii+V2S11+IKii11+IKiaS11+SKsi+K0.5s11+IKii11+IKia+IKsi

Boxplots and correlation plots were used to evaluate the fit associated with each model. Ideally a correlation plot of calculated vs. observed data should produce a slope of one (Fig. 4) and an R2 as close to one as possible, providing a visualization of the model’s fit. The residual boxplot provided a similar representation, where improvements in fitting of the models were evaluated based on decreases in spread and increased symmetric distribution around zero. The correlation plot produced by Eq. (10) (Fig. 4A) suggested that it was able to approximate the data fairly well. However, the boxplot produced a negative asymmetric distribution of the residuals (Fig. 5A). Refitting the kinetic parameters associated with Eq. (10) improved the slope of the correlation plot (Fig. 4B) and also improved the symmetric distribution of the residuals around zero (Fig. 5B). Equation (5) however improved both the slope and the R2 value for the correlation (Fig. 4C). A marked improvement in the symmetry and spread of the residual values was also observed (Fig. 5C). However, as previously mentioned Eq. (10) only relies on five kinetic parameters while Eq. (5), when expanded to describe DAPT interactions, contains 17. Thus, an increase in kinetic parameters might be viewed as over fitting of the data as models of greater complexity are known to produce improved fitting (Burnham & Anderson, 2002). To evaluate whether the improvement in fitting provided by Eq. (5) resulted from over fitting, Eqs. (5) and (10) were compared using the bayesian information criterion (BIC). BIC was developed to specifically penalize increasing complexity in model selection where a difference greater than ten is considered strong evidence against the higher value (Burnham & Anderson, 2002; Faraway, 2004). Not surprisingly when the BIC was used to compare Eq. (10) with the results produced by refitting the kinetic constants of Eq. (10) (Figs. 4A–4B), a significant improvement in the BIC value was observed (published values BIC = 597, refit kinetic parameters BIC 515). The decrease in BIC was attributed to a reduction in residuals without any increase in the complexity of Eq. (10). Examination of the BIC value produced by Eq. (5) (BIC = 448) also suggested a significant improvement over the fit achieved with Eq. (10) even though the number of parameters had increased by 12.

Figure 4 Global equation fitting to the experimental data.

Correlation plots of experimentally observed reaction rates vs. calculated values obtained using (A) Eq. (10) with the reported kinetic constants (correlation r = 0.968), (B) Eq. (10) with kinetic constants optimized in Excel (r = 0.972) and (C) Eq. (5) (r = 0.993).

Figure 5 Boxplot of the residuals associated with each data fitting.

(A) Residuals produced by Eq. (10) with the published values, (B) residuals associated with the Eq. (10) after it was refit to the data and (C) residuals associated with Eq. (5). Center lines show the medians; box limits indicate the 25th and 75th percentiles as determined by R software; whiskers extend 1.5 times the interquartile range from the 25th and 75th percentiles; outliers are represented by dots. Since Eq. (10) was not fit to background substrate concentrations, for A and B, n = 208, while for C (Eq. (5)) n = 230 sample points. This plot was generated using the web-tool BoxplotR (Spitzer et al., 2014).

The improper fit produced by Eq. (10) may relate to several factors, such as a lack of hill coefficients in the equation. Even though, individual hill coefficients were used in the initial description of the raw experimental data (Svedruzic, Popovic & Sendula-Jengic, 2013) they were left out of the simplified form of the equation ultimately published. Additionally, Eq. (10) only models C99-APP expression. The equation does not take into account normal cell line amyloid precursor protein expression levels, i.e., the substrate concentration is defined as the amount of vector used to produce the substrate and excludes endogenously produced substrates. However, neither of these things can be easily evaluated due to the confusing notation of the equation that includes terms that do not distinguish between binding and effect. In addition, the paper lacks a proper overlay of the experimental data with the model produced using Eq. (10).

Concluding Remarks

The information that can be obtained through proper kinetic modeling is invaluable for the understanding of processes at the molecular level. The way mechanistic equations are currently developed obscures the relationships between binding constants and effect on the enzyme. This lack of utility and obscurity has led to the marginalization of the enzyme kinetic field in a decade where tremendous amounts of money are being funneled into understanding biology at the molecular level. While the improper use of kinetic models in drug development and the study of biological processes may not be the primary cause of the problems plaguing the biological sciences (Wadman, 2013) it is contributing to them. This is exemplified by the development of DAPT as a γ-secretase inhibitor. Elan Pharmaceuticals, rather than doing a thourough characterization, used the standard IC50 method (Dovey et al., 2001). The value they reported (IC50 20 nM) suggested they only looked at DAPT’s effect at higher substrate concentrations (Burton et al., 2008). The results, more than a decade of research into a compound that holds very little promise of ever being useful in the treatment of Alzheimer’s disease. While DAPT never made it to clinical trials and our understanding of γ-secretase benifited from its study, the kinetic models used to describe DAPT’s effect potentially delayed appropriate insights by a decade. Indeed based on these inhibitors new classes of γ-secretase modulators are being developed as potential therapeutics for Alzheimer’s disease. However, characterization of their efficacy are once again relying on IC50s and EC50s (Mitani et al., 2012). The failings of γ-secretase inhibitors clearly demonstrate that when assessing the effect of compounds on γ-secretase activity it cannot be taken for granted that the effect will stay the same over varying substrate concentrations. The development of DAPT as a γ-secretase inhibitor only hints at the enormity of time, money and resources that have been lost as a result of the marginalization of enzyme kinetic in favor of simplified inhibition models or IC50s.

The failure of Alzheimer’s disease drug candidates have been attributed to many factors such as the initiation of clinical trials without proper insight into therapeutic mechanisms, improper design of the studies and a lack of mechanistic understanding of the disease itself (Becker et al., 2014; Schneider et al., 2014). To address these issues, Becker et al. (2014) have stated that the development of sound scientifically grounded mechanistic theories of disease progression needs to be a priority. While it is easy to agree with these ideas, the persistent use of inappropriate kinetic models, which mask more complex molecular interactions, will continue to obscure both disease mechanism and potential therapeutic intervention.

Supplemental Information

Supplemental Information 1 Gamma-secretase DAPT data fitting

Click here for additional data file.

Supplemental Information 2 Description of constants used in the fitting

Click here for additional data file.

Figure S1 Boxplot of the residuals associated with each data fitting

(A) Residuals produced by Eq. (10) with the published values, (B) residuals associated with the Eq. (10) after it was refit to the data and (C) residuals associated with Eq. (5). Center lines show the medians; box limits indicate the 25th and 75th percentiles as determined by R software; whiskers extend 1.5 times the interquartile range from the 25th and 75th percentiles; outliers are represented by dots. Since Eq. (10) was not fit to background substrate concentrations, for A and B, n = 208, while for C (Eq. (5)) n = 230 sample points. This plot was generated using the web-tool BoxplotR (Spitzer et al., 2014).

Click here for additional data file.

I would like to thank Dr. Svedružić for kindly sharing the raw data from his study on DAPT and γ-secretase interactions.

Additional Information and Declarations

Competing Interests

Author Contributions

The author declares there are no competing interests.

Ryan Walsh conceived and designed the experiments, analyzed the data, contributed reagents/materials/analysis tools, wrote the paper, prepared figures and/or tables, reviewed drafts of the paper.

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
