# Peer review of "Are improper kinetic models hampering drug development?"

_PeerJ, doi:10.7717/peerj.649_

## Round 0.1 · original submission · Major Revisions

As you will see, we got divergent opinions from the reviewers. In addition, I was informed by the Editorial Office that the present submission could actually have been a re-submission of a paper that was originally rejected. If such is the case, when responding to the reviewers reports, please also provide a detailed answer to the critiques raised against this first submission and explain how the present submission differs from it.

·

Basic reporting

This manuscript proposes new empirical equations to interpret complex enzyme inhibition data. The approach was applied to previously published beta-amyloid concentration-response data from HeLa cells treated with DAPT, a gamma-secretase inhibitor. The Author concluded that the new empirical equation led to a better description of the experimental data. He claimed that the improper fitting of complex enzyme inhibition data may led to wrong development of drugs.

Experimental design

The manuscript describes, in an elegant way, the derivation of the new empirical equations for complex drug-enzyme interaction data starting from traditional enzymatic models (competitive, non-competitive, mixed). The new empirical equations were applied to cell-based beta-amyloid production data described in a previous paper (Svedruzic et al, PLOS One 2013).

Validity of the findings

Author should note that data he used for testing the new empirical equations refers to the beta-amyloid1-40 peptide and not to the beta-amyloid1-42 peptide, the molecular species considered more neurotoxic and relevant for Alzheimer’s disease pathogenesis. Thus, the parameters derived by calculations may have limited pharmacological relevance.
Author should also note that different lines of Figure 2 refers to different concentrations of the C99 substrate of gamma-secretase (C-terminal fraction of amyloid precursor protein or CTF-beta), not to beta-amyloid precursor protein (APP).

Additional comments

While the rationale and mathematical derivation of the new empirical equations are well and elegantly described, I do not believe that the resulting data fitting was dramatically different from those presented in the original PLOS One paper. Most importantly, it should be pointed out that DAPT never entered studies in humans and this was for toxicity reasons mainly linked to lack of selectivity (interaction with Notch receptors, etc.). After the negative results of clinical trials with semagacestat (Doody RS et al. N Engl J Med 2013; 369: 341-350) and avagacestat (Coric V et al. Arch Neurol 2012; 69: 1430-1440), in which Alzheimer patients receiving the drugs declined cognitively more rapidly than those assigned to placebo, the gamma-secretase inhibitors have been abandoned as anti-Alzheimer drugs. Biochemically, the detrimental effects of gamma-secretase inhibitors on cognitive function have been ascribed to accumulation of C99, the substrate of the enzyme (Mitani Y et al. The Journal of Neuroscience 2012; 32: 2037–2050). So, the failure of DAPT and of other gamma-secretase inhibitors in Alzheimer’s disease cannot be ascribed, as suggested by the Author, to wrong estimation of the IC50 on the production of beta-amyloid1-40, but rather to a lack of selectivity (inhibition of Notch, N-cadherin, or EphA4) and on the intrinsic mechanism of action (accumulation of the gamma-secretase substrate C99 that, at high non-physiological concentrations, has amnesic activity). Thus, I believe that idea of the Author that “improper kinetic models hampered the development of DAPT or other clinical candidates in Alzheimer’s disease” is an overstated claim.

Reviewer 2 ·

Basic reporting

No

Experimental design

No

Validity of the findings

No

Additional comments

The manuscript submitted by the Ryan Walsh is very relevant in the current context of enzyme inhibition kinetics model for designing of new drug candidates for management/cure of various diseases. The manuscript will target a specific audience in the field of enzymology and will be a good reference document for researchers starting their enzymatic research work or building upon their current bio-catalytic work. References are appropriate and have been cited at apt places. I believe that this manuscript should be published with some minor edits as suggested below.
1. Better to replace ‘ic50’ by ‘IC50’.
2. To include one para for assay of γ–secretase with respect to predicted concentration of APP in vivo state.
3. To compare current analysis (Fig. 2) with classical IC50 and AC50 (that concentration of DAPT increases 50% γ–secretase activity at lower concentration range as compared to higher concentration range state, where determined IC50).
4. A thorough proof reading of manuscript will be required to eliminate typos (e.g. Figure 3. . Schematic) and grammatical mistakes to improve the quality of manuscript.
5. There are also some other mixed and uncompetitive types of inhibition (pure and partial), which would be discussed in this ms with references.
6. Check list of abbreviations that fulfill the criteria of minimum 3 times cited in the text.

7. Make sure that format and style should be according to the journal style.

---

## Round 0.2 · accepted · Accept

I guess you have satisfactorily answered the main criticisms of the reviewers.